# Scoping Review on Platelets and Tumor Angiogenesis: Do We Need More Evidence or Better Analysis?

**DOI:** 10.3390/ijms232113401

**Published:** 2022-11-02

**Authors:** Arianna Filippelli, Cinzia Del Gaudio, Vittoria Simonis, Valerio Ciccone, Andrea Spini, Sandra Donnini

**Affiliations:** 1Department of Life Sciences, University of Siena, Via A. Moro 2, 53100 Siena, Italy; 2Department of Medical Science, Surgery and Neuroscience, University of Siena, 53100 Siena, Italy; 3Azienda Ospedaliera Universitaria Senese, 53100 Siena, Italy

**Keywords:** tumor angiogenesis, platelets, tumor microenvironment

## Abstract

Platelets are an active component of the tumor microenvironment (TME), involved in the regulation of multiple tumor processes, including angiogenesis. They are generated rich in angiogenic factors in their granules to actively participate in the hemostatic process by megakaryocytes and further enriched in angiogenic factors by all components of the tumor microenvironment to control the angiogenic process because of their preferential relationship with the endothelial component of vessels. In recent decades, the literature has reported a great deal of evidence on the role of platelets in tumor angiogenesis; however, it is unclear whether the number or mean volume of platelets and/or their content and localization in TME may have clinical relevance in the choice and management of therapy for the cancer patient. In this scoping review, we collected and critically reviewed the scientific evidence supporting a close relationship between platelets, cancer, and angiogenesis. The aim of this work was to define the landscape of platelet-activated angiogenesis in cancer progression and analyze what and how much evidence is present in the last 20 years in the literature at both the preclinical and clinical levels, to answer whether platelets could be a useful determinant for analyzing tumor angiogenesis. In conclusion, this scoping review indicates that there is much evidence, both preclinical and clinical, but in the preclinical context, studies demonstrate the direct involvement of platelets in tumor angiogenesis; in the clinical context the evidence is indirect, though strong, and the indication of how and to what extent platelet content contributes to tumor angiogenesis is lacking. So, do we need more evidence or better analysis? More molecular and quali-quantitative data is needed to translate the results obtained in preclinical studies into the clinical setting. This information about platelets, if correlated with tumor type and its biology, including tumor vasculature, type of angiogenesis, and patient characteristics (age, sex, comorbidities, drug treatments for chronic diseases) could be an important pa- rameter for correlating platelet biology to angiogenesis, for personalizing cancer therapy, and for clinical prognosis.

## 1. Introduction

It is now well established that cancer cells co-exist within a complex environment with stromal cells and depend for their growth and dissemination on tight and plastic interactions with components of the tumor microenvironment (TME), including platelets and endothelial cells (ECs) of the blood and lymphatic vasculature. Platelets in collaboration with the other TME components drive a chronic inflammatory, immunosuppressive, and hypoxic pro-angiogenic niche that promotes the dissemination of cancer cells and limits the effects of various therapeutic interventions, including immunotherapy [1,2,3].

Pathological angiogenesis is mainly driven by an imbalance between pro-angiogenic and antiangiogenic signaling in the TME. Key pro-angiogenic factors include vascular endothelial growth factor-A (VEGF-A), and fibroblast growth factor-2 (FGF-2), interleukin (IL)-8, IL-10, and prostaglandin E2 (PGE2) [4,5,6,7,8]. These molecules become abundant in the TME and overwhelm angiostatic signals, thereby inducing a pro-angiogenic switch [9].

Research over the past decades mainly focused on the role of cancer cells in orchestrating the formation of new blood vessels from pre-existing vessels to favor tumor outgrowth, facilitate metastasis, and contribute to immunosuppressive TME, while the contribution of the other components has been more neglected [10,11,12,13]. With reference to platelets, despite the fact that they have been attributed a role in tumor angiogenesis, to date, in the era of personalized medicine and liquid biopsy, it is unclear whether platelet accumulation in TME, the number of platelets in bloodstream and their contents may have clinical relevance [14].

Platelets represent one of the smallest cellular elements that populate the blood stream. These megakaryocyte derivatives are produced in the bone marrow and spread into blood stream. Platelets exhibit a brief lifespan (7–10 days); however, they are recognized as one of the most important and polyhedric cellular elements of blood [15,16,17]. Circulating platelets exhibit heterogeneous age, dimension, and number; this variability allows them direct involvement in various physiological and pathological processes, making these blood elements leading players in scientific investigation in recent decades [18,19,20,21]. The well-known mechanisms of rapid aggregation and plug formation at the site of injury, during physiological hemostasis, together with the pivotal role played at the onset of pathological thrombosis, are enriched by several varied functions recently discovered, which make platelets major effectors in several additional functions, including cancer progression [16,22,23,24,25,26,27] (Figure 1). The key factor in the multifaced role of platelets lies in their ability to respond to signals from the endothelium, such as the gasotransmitters nitric oxide (NO) and hydrogen sulfide (H2S) [5], from the circulating cells or other blood components, by maintaining a continuous crosstalk with their environment through cell surface receptors and adhesion molecules (i.e., integrins, selectins, toll-like receptors, transmembrane receptors, immunoglobulin superfamily receptors, tyrosine kinase receptors, lipid receptors, and others) [28,29,30].

Platelets during their lifetime develop intricate and continuous crosstalk, through which they also exhibit the ability to modulate their environment. Platelet adhesion to the extracellular matrix (ECM), through exposed collagen and platelet glycoprotein receptors, triggers platelet shape changes and the release of bioactive mediators from storage granules (i.e., α-granules, dense granules, and lysosomes), bioactive lipid products, and extracellular vesicles [18,28,29,31]. Behind the bioactive trafficking activated by platelet secretion are numerous coagulation and growth factors, chemokines, cytokines, eicosanoids, microbicidal agents, and RNAs [14,29].

In TME, platelets release circulating angiogenesis-related factors, including VEGF, platelet-derived growth factor (PDGF), FGF, and metalloproteinases (MMPs); in addition, they promote the expression of proangiogenic factors by tumor cells [32]. Platelet alpha (α) and dense (δ) granules contain one of the largest stores of angiogenic and mitogenic factors (VEGF, PDGF, FGF-2, MMPs), antiangiogenic factors (endostatin, thrombospondin-1, plasminogen activator inhibitor-1, and angiostatin), chemokines (platelet factor 4, PF4, RANTES), and cytokines (IL-1β), which markedly affect vascular biology [33,34,35,36]. On the other hand, leaky tumor vasculature allows platelets to directly meet the tumor, and tumor cells activate platelets, leading to platelet aggregation and the release of platelet-derived growth and proangiogenic factors, which in turn, in a vicious circle, support tumor growth and angiogenesis [35,37,38].

The platelet storage compartment, consisting of α-granules, dense granules, and lysosomes, is highly regulated. Controversial evidence indicates the possibility that α granules are divided into distinct subpopopolations and release by a stochastic process, or, conversely, that they are differentially regulated in the release of angiogenic mediators in response to platelet agonists [39,40,41,42,43].

This mechanism may explain how platelets contribute to active angiogenesis early in wound healing and inhibit angiogenesis in the late stages of inflammation. Similarly, the ability of platelets to store either proangiogenic or antiangiogenic factors suggests an involvement of platelets in tumor progression or dormancy [37,39,44,45,46].

Some platelet-selective proteins, such as PF4 and thrombomodulin, are synthesized by several cells, including megakaryocytes, and concentrate in platelets in α-granules [22]. Other proteins, such as VEGF and FGF-2, are either synthesized or taken up by platelets and concentrated in α-granules [37]. In human-tumor-xenograft-bearing mice, the synthesis of PF4, an angiogenesis suppressor, is upregulated in the host (murine) bone marrow megakaryocytes [47]. Further, the levels of VEGF and FGF-2 in circulating platelets is increased in mice bearing human-tumor xenografts compared with tumor-free controls [48], suggesting that tumors may affect the net proteomic content in platelets regulating how the proteins are redistributed and secreted and that platelet levels of angiogenesis-related proteins may be higher than plasma and serum levels. Additionally, endothelial cells in tumor tissues can capture platelets by augmented tissue factor (TF) expression [1,49]. Thus, TF finally generates thrombin and activates platelets via protease-activated receptor 1 (PAR-1), which culminates in local platelet settling and growth factor release [1,50,51].

The close and constant interactions between platelets and cancer cells converge in a pro-coagulant milieu in the tumor site [1,32,52]. In 1998, Pinedo and Folkman stated that, presumably, increased platelet adhesion to the tumor vasculature and the subsequent platelet activation are regulated by tumor cell stimuli and may differ for each tumor type. However, they presumed that the number of platelets adhering and (partially) activated was critical in determining these effects on tumor angiogenesis and that platelet population heterogeneity and differences in platelet number could cause different effects on the tumor vasculature [35]. Twenty years after this seminal study, as integral components of the TME, there is clear experimental evidence that platelets contribute to the remodeling of the tumor vasculature through several mechanisms (endothelial cell proliferation [53], the recruitment of pericytes and bone-marrow-derived cells [54], in addition to vessel stabilization [55,56], mediated by both platelet angiogenic factor releasate and platelet ligand–receptor interactions [57,58] (Figure 1). Less clear is the contribution of platelets in the clinical setting. Of note, anti-platelet-aggregation drugs may block the differential release of angiogenic regulators from platelets, thereby regulating platelet-dependent tumor angiogenesis [59,60,61]. Specifically, platelets are reported to be involved in both cancer angiogenesis and vascular normalization, processes closely related to cancer progression and drug distribution, respectively. Interestingly, several studies in mouse models of solid tumors suggest that targeting the vessel stabilization function of platelets in tumors may improve the intratumoral release and efficacy of chemotherapeutic drugs [62,63,64]. Collectively, evidence from the literature suggests that, in different cancers, platelets affect differentially cancer pathogenesis [65,66] and progression [67]. Therefore, the pattern of angiogenic factors in platelets may be a valuable biomarker for the personalized combinatorial strategies of antiangiogenic drugs with chemo/target and immune therapies [18,68].

In this scoping review, we aimed to collect and critically review the scientific evidence supporting a close relationship between platelets, cancer, and angiogenesis; in particular, the focus of this work was to define the landscape of platelet-activated angiogenesis in cancer progression, to answer whether platelets may be a surrogate for tumor angiogenesis or a useful factor in analyzing tumor angiogenesis.

## 2. Materials and Methods

### 2.1. Study Design

This is a scoping review of the literature and protocol was registered in an OpenScience Framework network (DOI 10.17605/OSF.IO/R53AP updated on 16 September 2022).

### 2.2. Literature Search

We searched PubMed and ISI Web of Science databases for retrieving articles of interest. Articles that were published from January 2000 to July 2022 were considered suitable for inclusion. The search string was composed of three sets of keywords respectively related to the following concepts: “platelets”, “angiogenesis”, and “cancer” (the full strings for both databases are reported in Appendix A). Snowballing search was also conducted to retrieve additional papers of interest by examining the references cited in the included articles that were retrieved from the search strategy.

### 2.3. Eligibility Criteria

All in vitro and in vivo preclinical studies, as well as clinical trials and observational studies based on platelets that reported evidence on angiogenesis, were selected (Figure 2).

Preclinical studies had to evaluate at least one of the following topics: (1) the association between tumor angiogenesis with platelet count in plasma/platelet accumulation in TME; (2) the effect of anti-angiogenic/proangiogenic factors in circulating/TME-associated platelets on cancer cells or in vivo tumor models; (3) the association between platelets and tumor growth/progression/metastasis or responsiveness to anti-angiogenic drugs alone or in combination with chemo- or immune-therapy in cancer cells or in vivo tumor models.

Clinical studies needed to evaluate: the association between platelets (plasma/serum) and the efficacy of anti-angiogenic drugs alone or in combination with chemo- or immune checkpoint inhibitors.

Studies with either no abstract or no full text available were excluded.

### 2.4. Study Selection

Three authors (AF, CDG, and VS) screened all titles and abstracts of the references retrieved. Potentially relevant studies were further assessed through an examination of the full texts. The reviewers worked independently, in parallel, and blinded to each other. Disagreements between the two reviewers were solved through discussion with a third author (SD).

### 2.5. Data Extraction

The following information was extracted from the included studies where available:(a)Platelets (count, distribution width, volume)(b)angiogenesis (endothelial cell tube formation, endothelial cell proliferation, endothelial cell migration, microvessel density, angiogenic growth factor concentration)(c)as for preclinical studies, tumor progression (volume, proliferation, metastasis, epithelial mesenchymal transition (EMT), stemness, invasiveness)(d)as for clinical studies, clinical outcomes (overall survival, progression-free survival, objective response rate).(e)antiangiogenic drugs (type)

A descriptive, qualitative synthesis of results was provided.

## 3. Results and Discussion

### 3.1. Literature Search Results

A total of 1173 studies were retrieved from PubMed and ISI Web of Science (Figure 2). The screening of titles and abstracts allowed the selection of 197 potentially eligible studies. Among them, a total of 24 studies fulfilled the eligibility criteria and were included in the review. No further studies were retrieved through a snowballing search. Overall, we found seven preclinical studies on cells [69,70,71,72,73,74,75], seven preclinical studies on in vivo tumor models [54,76,77,78,79,80,81], and ten clinical studies [71,75,82,83,84,85,86,87,88,89,90]. Two articles presented both preclinical and clinical evidence on platelets and tumor angiogenesis and were analyzed in both the results dedicated to preclinical and clinical evidence [71,75].

### 3.2. Preclinical Studies on Cells

Related to the analysis of the association between platelet content and tumor angiogenesis, seven articles were retrieved from literature databases (Table 1; [69,70,71,72,73,74,75]). There are several tests that can be used to study angiogenesis in vitro, including measuring the ability of the endothelium to form capillary-like structures. In all articles retrieved, tumor angiogenesis in vitro was also studied by the endothelial tube formation assay, a validated test for the study of angiogenesis, although limited in reflecting the complexity of TME. We therefore selected this assay as a parameter of pro-angiogenic related-platelet function (Table 1). Four of the seven articles studied the pro-angiogenic imprinting that lung cancer cells give to platelets [69,70,71,72]. Wu et al. correlated the whole platelet releasate with increased capillary formation and the migration of human umbilical vein endothelial cells (HUVEC) [69]. The authors reproduced in vitro the tumor-induced activation of platelets mixing and manipulating non-small cell lung cancer (NSCLC) cells, A549, with platelets from healthy volunteers. The activation and “education” of platelets (platelets with tumor-derived RNS and proteins) from NSCLC converged on high increased capillary formation and migration by endothelial cells [69]. Further, the observation of increased levels of pro-angiogenic factors in platelet releasate confirmed the pro-angiogenic behaviors of NSCLC cells on activated platelets [14,69]. Li et al. reached the same conclusion [72]. Co-culturing lung carcinoma cells, H1975, with platelets improved the pro-angiogenic functions of HUVECs in terms of proliferation, migration, and tube formation. Consistently, Wang et al. corroborated this evidence, observing that platelet releasate, derived from NSCLC patients, not only increased cancer cell proliferation, but also impaired the sensitivity to the drug cisplatin. This mechanism was correlated by the authors to a pro-angiogenic profile in platelet releasate and to their capacity to increase tube formation in pulmonary endothelial cells [70]. The relationship between NSCLC cells and intra-platelet levels of VEGF, TSP-1, and net platelet angiogenic activity (NPAA) was deeply investigated by Yao and his group [71]. In their study, they compared the platelet releasate from healthy donors and NSCLC patients, profiling the pro-angiogenic factors and cytokines of each sample from a molecular and clinic point of view. They demonstrated that the platelet releasate derived from NSCLC patients showed high NPAA activity, investigated as an increase in endothelial cell tube formation ability [71].

Very few reports correlated platelets, angiogenesis, and cancer progression in other solid tumors, including breast, ovary and glioblastoma [73,74,75]. Jian et al. investigated breast-cancer-cell-induced endothelial tube formation in vitro in the presence of PAR-1- or PAR-4-stimulated platelet releasate. They showed a significantly increase in capillary-like tube formation by endothelial cells, not only in the presence of platelet-releasate induced by PAR-1 or PAR-4, but mainly in the presence of both breast cancer models, MCF-7 and MDA-MB-231. The cancer cells triggered capillary-like tube formation by endothelial cells that was further enhanced by PAR1/4-platelet releasate, rich in pro-angiogenic factors. Of note, the PAR-1- and PAR-4-induced platelet releasate similarly enhanced, by around 50–70%, the cell proliferation of MCF-7 and MDA-MB-231 cells [73]. Authors concluded that the platelet releasate increased breast cancer cell proliferation through VEGF–integrin cooperative signaling, and the pro-angiogenic factor-rich platelet releasate enhanced cancer-cell-induced angiogenesis more markedly.

Platelets extracted from healthy volunteers were also used to investigate the role of the platelet releasate in ovarian cancer. Aiming to demonstrate that concentrations of metformin, used in diabetic patients, might reduce the imprinting of platelets on both endothelial and cancer cells, Erices et al. tested the platelet-conditioned medium on ovarian cancer lines (SKOV3, UCI101) and endothelial cells (EA.hy926, HUVEC). They created a complex system of interplay between these three elements, co-colturing ovarian cancer cells with platelets in the presence or absence of metformin, and after 24 h, the conditioned media was used to evaluate angiogenesis in EA.hy926 cells and the balance of pro-angiogenic factors released from both the SKOV3 and UCI101 ovarian cancer cell lines. An increase in pro-angiogenic factor released and an increase of up to 50% in capillary-like tube formation was reported by the authors, correlated with a higher increase of cancer cell migration. Of note, a 24-h incubation of platelets in culture medium in the absence of cancer cells did not significantly increase angiogenesis, demonstrating that the increase in the balance of pro-angiogenic factors originated from the cancer cells upon contact with platelets [74].

The role of platelets was also investigated in glioblastoma multiforme (GBM) progression. The manuscript of Di Vito et al. reported that platelet releasate strongly enhanced the ability to form a capillary-like tube in glioblastoma-derived endothelial cells (GECs). In this study, authors collected platelets from GBM patients and healthy volunteers at the same time and isolated endothelial cells from glioblastoma tissues to test the biologic relevance of platelets on the angiogenesis of GBM and to screening GBM patients for anti-angiogenic therapy. The ability of GECs to form capillary-like structures was influenced by the presence of the ADP-induced releasate from the platelets of either healthy controls or glioblastoma patients. The authors recognized that ADP may influence the formation of capillary-like structures per se; however, the capacity to stimulate the angiogenesis of platelet releasate from glioblastoma patients was greater than that from healthy individuals, and they demonstrated that the pro-angiogenic effect of ADP-induced platelet release on GECs was mediated by VEGF [75]. In the other articles, platelet release was achieved with nonangiogenic stimuli.

### 3.3. Preclinical Studies on In Vivo Tumor Models

Evidence that platelets can impact tumor angiogenesis has been reported from several studies performed on mice bearing human-tumor xenografts or syngeneic B16-F10 tumor models [54,76,77,78,79,80]. In our analysis, seven articles were retrieved that demonstrated the strong relationship between platelets and tumor angiogenesis (Table 2). Four out of seven studies [54,76,78,79] demonstrated a decrease in microvessel density in syngeneic B16-F10 melanoma and human ovarian xenografts after platelet depletion through an antiplatelet antibody directed against mouse glycoprotein 1bα (GPIbα), administered in combination with tumor implantation. In addition, Li R. et al. observed a significant reduction in Ang-1 and an increase in Ang-2 levels in platelet-depleted B16-F10 melanoma compared to control tumors [76].

With an opposite experimental approach, Feng et al. and Yuan et al. also investigated platelet infusion in syngeneic B16-F10 melanoma and human ovarian xenografts, demonstrating an increase in tumor microvessel density [54,79]. In particular, a study by Yuan et al. first investigated platelet count in ovarian tumor patients and then analyzed angiogenesis in human SKOV3 tumor xenografts. It showed that ovarian carcinoma patients had a significantly higher platelet count compared with patients with borderline cystadenoma and cystadenoma, and it showed that patients with carcinoma expressed significantly more VEGF and microvessel density compared with the other two groups [79]. To confirm the role of platelets in tumor angiogenesis, Zaslavsky et al. studied the function of TSP-1. Because TSP-1 is a potent angiogenic inhibitor, they quantified tumor angiogenesis in murine B16-F10 melanoma and human Lewis lung carcinoma isolated from mice Tsp-1 deficient (Tsp-1^−/−^) and wild-type platelets [77]. Tumors isolated from mice Tsp-1^−/−^ had substantially higher microvessel density than tumors isolated from mice with wild-type platelets for TSP-1. Even tumor volume was higher in mice injected with Tsp-1^−/−^ platelets. Only two out of seven articles [80,81] showed a contrary result compared to the canonical mechanisms of tumor angiogenesis. In the study of Martini et al., researchers used Bcl-x Ptl20/Ptl20 mice, wherein platelet counts were reduced by ±25%, and they demonstrated that the B16-F10 melanoma in the Bcl-x Ptl20/Ptl20 mice contained significantly more vasculogenic mimicry structures than their wild-type counterparts. In addition, there were no differences in tumor size (volume and weight) between the two groups [80]. Consistently, Brockmann et al. showed that tumor volume and microvessel density analyzed in nude mice wherein G55T2 glioblastoma cells were implanted did not differ between animals with a normal and a low platelet count [81].

### 3.4. Clinical Studies

We retrieved 11 clinical studies, including the studies by Di Vito et al. and Yao et al. in which preclinical evidence was reported [71,75], that showed a correlation between platelets and tumor angiogenesis (Table 3 [71,75,82,83,84,85,86,87,88,89,90]). Ten studies were observational and retrospective, and only one clinical trial phase I was collected [85]. In 4 articles out of 11 the correlation was investigated in breast cancer [85,86], in 3 in gastrointestinal cancer [82,88], in 2 in lung cancer [71,87], and only in 1 each in pancreatic cancer [87], in glioblastoma [75], and in chronic myeloid leukemia [90]. The involvement of platelets in tumor growth was investigated as platelet count and/or median platelet volume (MPV). In five studies, the platelet count in patients with tumors was higher than in healthy controls, as much as the levels of MPV. In addition, the correlation between the platelet count and the stage of the tumor was reported [83,84,85,87,88]. The involvement of platelets in tumor angiogenesis was investigated as an analysis of platelet’s angiogenic factors, the most studied of which was VEGF. It has been analyzed in ten different tumors. In seven of them, it was related to the tumor stage [71,82,83,84,85,86,87]. In all cases, VEGF levels in patients with cancer were higher than in the healthy controls [71,82,83,84,85,86,87]. The same results were found for other platelets’ angiogenic factor. In two studies, researchers obtained and analyzed the platelet-rich plasma (PRP) or the platelets pellet [71,86]. Then the values of VEGF, IL-6, TSP-1, PF4, TGFβ, and PDGFβ were divided by the platelet count to show that the levels of angiogenic factors have been released by platelet [71,83,86,87]). McDowell and colleagues studied whether the platelets of patients with tumors were functionally altered compared with the normal control through a determination of the increase in the concentration of VEGF or TSP-1 that has been released due to thrombin or TF stimulation [86]. The results have shown that the platelets from patients with cancer released more VEGF with thrombin stimulation, TF stimulation, or time. There was a relationship among the different markers. Kim et al. found a strong correlation between VEGF and IL-6, and the correlation was also found between platelet microparticles (PMP) and IL-6 [84]. Otherwise, platelet count was related to PMP and IL-6. In addition, Han et al. divided cancer patients into groups according to their tumor stage, age, and hormonal receptor expression, and they found that the level of VEGF, PDGFβ, and TGFβ in platelets was associated with clinical characteristics [83]. The study of Mayer et al. evaluated an antiangiogenic therapy (vandetanib, inhibitor of VEGF receptor 2, epidermal growth factor receptor, and rearraged during transfection receptor (RET)) in metastatic breast cancer, and the end point included the evaluation of platelet proteins [85]. The proteomic analyses showed changes in the platelet content of the angiogenic factor, including VEGF and PF4, with exposure to therapy. They thought that the changes in the platelets’ proteome help as pharmacodynamic markers of angiogenesis inhibition [85]. The strongest correlation between platelet angiogenic activity in cancer was also shown by Yao et al. in their study. They investigated VEGF, TSP-1, and the net platelet angiogenic activity (NPAA) (data not shown in Table 3) in the platelets of cancer patients compared to healthy controls [71]. To evaluate the NPAA, they tested the in vitro effect of platelet lysates from patients on the formation of HUVECs’ capillary-like structure through capillary-tube-formation assays. Median values of NPAA and VEGF in cancer patients were significantly higher than in healthy controls. Patients with high levels of NPAA were more likely to exhibit aggressive clinical pathological features, which were associated with poor overall survival. In addition, there was a significant correlation between NPAA or VEGF and tumor MDV in the cancer patients evaluated by CD34 immunohistochemical staining [71]. Another study that determined a correlation between platelet factors and MVD using antibodies against CD34 was that by Tokyol C. et al. [89]. They showed that the platelet counts were positively correlated with tumor size, and platelet counts were higher in patients who had major MVD (grade 3) [89].

The only non-solid tumor included in our analysis is chronic myeloid leukemia (CML), in this case Nafady et al. found that the percent of PMPs was significantly higher in patients with CML at diagnosis compared with healthy controls [90]. In this study, patients with CML had significantly higher angiogenesis parameters, measuring microvascular density and a number of different parameters related to the size and shape of microvessels, compared with controls [90]. The glioblastoma (GMB) was another tumor in which was found a direct correlation between platelet-released VEGF and overall survival. As reported above, the correlation of patients’ platelets with tumor and angiogenesis has also been proven in vitro (Table 1) [75].

## 4. Discussion

The crosstalk of platelets with TME components in tumor progression is a milestone in cancer research [1,2,3,37]. The present review aims to provide a comprehensive overview about the role of platelets in cancer progression, mediated by the modulation of tumor angiogenesis. The contribution of TME components in the processes of tumor progression has received much attention in recent years. Among the components of the microenvironment, platelets, because of their rich content of growth factors and cytokines, have also been of interest both in their relationship with tumor cells and with TME cells, including the endothelium. The evidence that platelets are rich in angiogenic factors and thus might play an important role in orchestrating tumor angiogenesis is from the last century. This scoping review brought together the work of the past 20 years on the topic of correlation between platelets and tumor angiogenesis to analyze what and how much evidence had been gathered. Interestingly, we found little evidence showing that platelets do not contribute to tumor angiogenesis process [80,81,91,92]. We think it unlikely that this is related to our search strategy (two large literature databases and an extensive search) but perhaps to publication bias. Therefore, we recommend that further researchers who intend to perform a systematic review on this topic also look for available evidence in the gray literature.

Our scoping review shows an intermittent interest in the pathophysiological role of platelets, with studies spread over a long-time span, about two centuries. The tumor type most studied in relation to platelets in in vitro studies is lung. The role of the lung as a site of megakaryocytes may justify the greater attention to lung cancer models than others (Table 1). The studies reviewed report the direct relationship between cancer, angiogenesis, and platelets with the aim of assessing the modulation and imprinting that platelets give to the tumor vasculature and subsequent changes in cancer behavior. Many other similar studies, which supported the involvement of platelets and their content or releasate in angiogenesis and cancer progression were excluded, because the correlation between the increased level of proangiogenic factors in platelet releasate and angiogenesis-supported cancer progression was indirectly observed and/or theoretically demonstrated by the authors, but no experimental data were presented that took these three elements into account. The manuscripts retrieved and included in our revision describe a clear association between platelet accumulation and angiogenesis in different cancer types (i.e., mesenchymal and epithelial ones), investigated as endothelial cell proliferation, migration, and the capability of forming capillary-like structures [69,70,71,72,74,75] (Figure 3).

The preclinical in vivo data, which covered heterogeneous cancer models, reinforce the observation reported in in vitro assays. The depletion of platelets in mice models is strongly correlated with a reliable decrease of microvessel density (MVD) in xenograft tumor mass (Table 2). In melanoma, lung, and ovarian cancer mice models, the platelet depletion and the subsequent decrease of MVD is negatively correlated with the tumor volume [77,79] and tumor weight [78]. An analysis of the main angiogenic factors (i.e, VEGF and Ang1–2) is only partially conducted, strongly supporting the role of platelets in angiogenesis and tumor growth.

Although in the literature retrieved from our research, it emerges that platelets have a positive impact on tumor angiogenesis, a process that is instrumental to tumor growth; there is also evidence in the literature that platelets have no impact or have a negative impact on both tumor cell proliferation [93,94,95,96] and tumor growth in mouse models of solid tumors [81,97], suggesting that platelets affect the tumor microenvironment (TME) through many mechanisms, and the resulting biological effect can vary greatly. Important determinants of the mechanisms and functional consequences activated by platelets in TME appear to be the type of cancer cells and platelet biology, including platelet reactivity and activation status, but also age and turnover rate [98]. Interestingly, in B16-F10 melanoma Bcl-x Ptl20/Ptl20 mice, the decreased number of platelets shows a significant increase in vascular mimicry [80], suggesting that the decrease of platelets may activate other compensatory mechanisms, requiring further analysis [10].

In clinical studies analysis, we mainly collected observational and retrospective studies (only one clinical trial phase I was retrieved [85]). The studies underly the close and reproducible correlation between platelet number or volume and cancer stage and patient prognosis (Table 3). Consistently, the levels of angiogenic factors, such as VEGF, IL-6, TSP-1, PF4, TGFβ, and PDGFβ were linked not only with the platelet count but also with cancer stage and patient prognosis [71,75,82,85,86,87,88,89,90]. However, clear evidence on if and how tumor platelet content correlates with tumor angiogenesis in cancer patients is still not present. Indeed, in cancer patients, the accumulation of platelets in bloodstream and the increase of angiogenic factors in tumor platelets are not clear evidence that platelets contribute to tumor angiogenesis and might be only an artifact or a biomarker of malignancy, instead of tumor angiogenesis.

In support of the active involvement of platelets in tumorigenesis (including tumor angiogenesis), there are numerous preclinical and clinical studies on the preventive and anticancer effect of antiplatelet agents, such as low-dose aspirin and P2Y12 purinergic receptor antagonists, such as clopidogrel. Since the seminal study of Algra and Rothwell in 2012 [99], numerous clinical data have confirmed the cancer-preventive role of antiplatelet drugs [100,101], and, more interesting for our scoping review, preclinical data have demonstrated the antiangiogenic impact of these drugs [102,103,104]. Although the targets of antiplatelet agents (COX for aspirin and purinergic receptor P2Y12 for clopidogrel) are also on endothelial cells, there are strong arguments [105,106] indicating that these drugs, especially aspirin, do exert their anticancer effects, and in turn probably their antiangiogenic effects, through platelet inhibition. Of note, the positive effect of aspirin as an antiplatelet on the risk of cancer occurrence, particularly colorectal cancer, is only evident in the young population. In the elderly population, the use of low-dose aspirin for primary prevention, in the ASPirin in Reducing Events in the Elderly (ASPREE) clinical trial, was associated with an increased mortality rate, largely attributed to higher cancer mortality [107] and highlighted a possible negative effect of aspirin on cancer progression [108]. The authors of this study conclude that molecular and genetic evidence of cancer in the elderly could justify the potential negative impact of aspirin in this specific age group. Overall, therefore, the impact of platelet activation in tumorigenesis is closely related to the microenvironment in which they act. The data collected and analyzed in this review support the key role played by platelets in TME in sustaining cancer progression, also through the modulation of the tumor vascularization process, and support its role as a potential clinical marker. Further, in recent years the presence of platelets as well as megakaryocytes has been demonstrated in the stroma of several tumor types, both in experimental mouse models of solid tumors and in human specimens, and their presence, due to localization on the invasive front of tumors or near vessels, could be indicative of an aggressive phenotype of tumors [109,110,111]. Of note, however preclinical data have in part been corroborated by clinical evidence, and this is mostly derived from observational, retrospective work, and to date in clinical practice, there is no indication of the use of platelets (number and TME accumulation) or their content as a prognostic or predictive biomarker for cancer patient management.

One of the main limitations to their use as a prognostic or predictive marker in oncology that makes their real applicability complex is their involvement in numerous pathophysiological processes [16] and their heterogeneity [112], associated with homeostatic events or the onset of a variety of diseases. The involvement of platelets in numerous pathologies that might potentially coexist in the cancer patient makes it difficult to identify the platelet-related parameter(s) to be evaluated as prognostic or predictive of response to anticancer drugs, including antiangiogenic drugs. On the other hand, today, innovative approaches and advanced technologies may be a valid support for studying platelets, not only as a secondary parameter in clinical trials, but also as rapid, simple, and valid biomarkers to be evaluated in cancer patients. Liquid biopsy follows by transcriptomic and proteomic analysis might be valid innovative approaches to study the tumor features from a platelet point of view [18,113,114]. This option can also simplify patient management, choice of treatment regimens, and define platelets as a reflex profile of TME and cancer characteristics. A more in-depth and constant scientific investigation needs to be conducted on the correlation introduced in this work, not only in the oncology field, but also carried out with other pathophysiological conditions, to profile patients in a non-invasive methodology [115].

Collectively, this scoping review brought together preclinical and clinical evidence from the past 20 years between platelets and tumor angiogenesis. Their easy handling, together with the large amount of biological information carried in their granules, contrasts with the neglected role that platelets play in clinical applications in oncology. However, their high heterogeneity and participation in important processes besides the hemostatic process currently makes the use of the knowledge gained about the contribution of these elements complex. In the omics analysis of tumors, however, these elements could assume an important role that requires further investigation

## 5. Conclusions

In conclusion, our descriptive review of the literature on the proangiogenic contri- bution of platelets in the tumor context indicates that there is much evidence, both pre- clinical and clinical, but, while in the preclinical context, studies demonstrate the direct involvement of platelets in angiogenesis, either by a mechanism of platelet/tumor cell and platelet/vascular cell interaction or by a mechanism of platelet activation and platelet factor release in the TME; in the clinical context, the evidence is indirect, and the indication of how and to what extent the platelet content contributes to tumor angiogenesis is lacking. Of note it is worth highlighting that some preclinical evidence in in vivo models of solid tumors is very strong [61,62,66,76].

We also know that the use of antiangiogenic drugs such as bevacizumab, a mono- clonal antibody directed toward VEGF, and tyrosine kinase inhibitors, such as sunitinib and sorafenib, pan-inhibitors of receptors for mitogens including that for VEGF, affected platelet biology, but there is no evidence on whether they modulate their pro-angiogenic role [116]. Bevacizumab has no effect on platelet aggregation or adhesion to the vascular endothelium, although the data are conflicting [117]. The clear evidence is that it is taken up by platelets and neutralizes the VEGF contained in granules, probably contributing to inhibiting tumor angiogenesis. Sunitinib and sorafenib, on the other hand, while not inhibiting platelet adhesion to the vascular endothelium, do inhibit platelet aggregation, accumulate there, and are responsible for hemorrhagic phenomena in patients with cancer. There are no studies directly correlating this aspect (hemorrhage and thrombocytopenia) with the inhibition of the pro-angiogenic role of platelets; however, they are suggestive of that.

So, do we need more evidence or better analysis? The answer is that more analysis is needed to be able to translate the results obtained in preclinical studies into the clinic, such as, purely by way of a non-exhaustive example, the quali-quantitative evaluation of the presence of extravascular platelets in the tumor mass, their isolation from the TME, and RNA-profile sequencing, and in circulating platelets, in addition to the number, the analysis of the transcriptome and the proteome profile.

A completely neglected aspect in the articles retrieved from our research is an analysis of platelet subpopulations [118,119]. Platelets are heterogeneous, and several platelet subpopulations with distinct functions have been identified and characterized, and there is recent evidence of their distinct contribution to tumor progression [120]. In this context, tumor-educated platelets (platelets containing tumor-associated mRNA and proteins) have been identified as a potential platform for blood-based liquid biopsies for cancer [121]. How platelet subpopulations change in cancer and how these changes impact their transcriptomic and proteomic content, and their biological functions (including angiogenesis) is unknown but likely of important significance to liquid-biopsy utility. The latter aspect requires further study.

Overall, the qualitative–quantitative and molecular information on platelets, when correlated with tumor type and biology and patient characteristics (age, sex, comorbidi- ties, drug treatments for chronic diseases), could be important parameters for correlating the role of platelets in different types of angiogenesis to personalize cancer therapy and clinical prognosis.

## Figures and Tables

**Figure 1 ijms-23-13401-f001:**
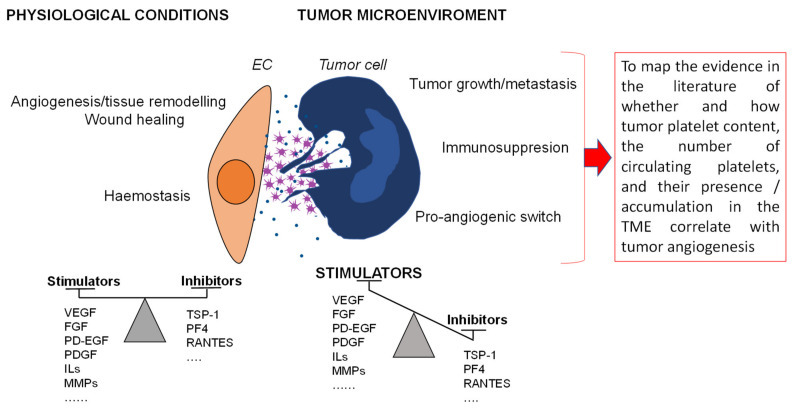
Representative scheme of the supporting role of platelets in tumor progression. Platelets play an important physiological role in hemostasis and wound repair and, in the tumor microenvironment, interacting directly with the vascular and tumor components support through the accumulation and release of growth and angiogenic factors tumor progression. 
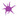
 Platelets; growth/inhibitory factor.

**Figure 2 ijms-23-13401-f002:**
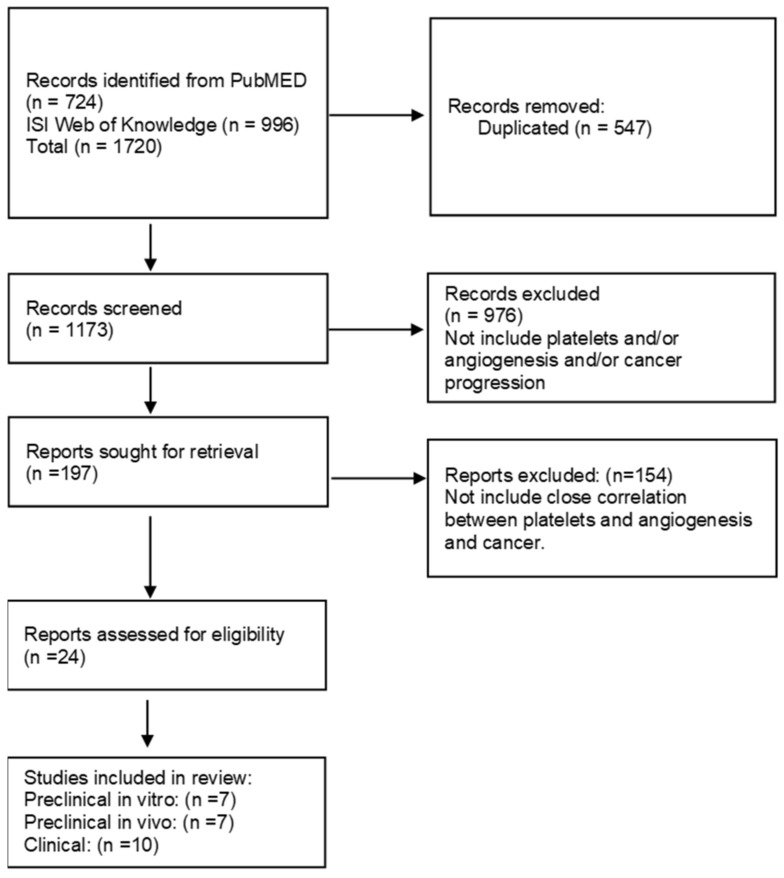
Flow diagram of screened records.

**Figure 3 ijms-23-13401-f003:**
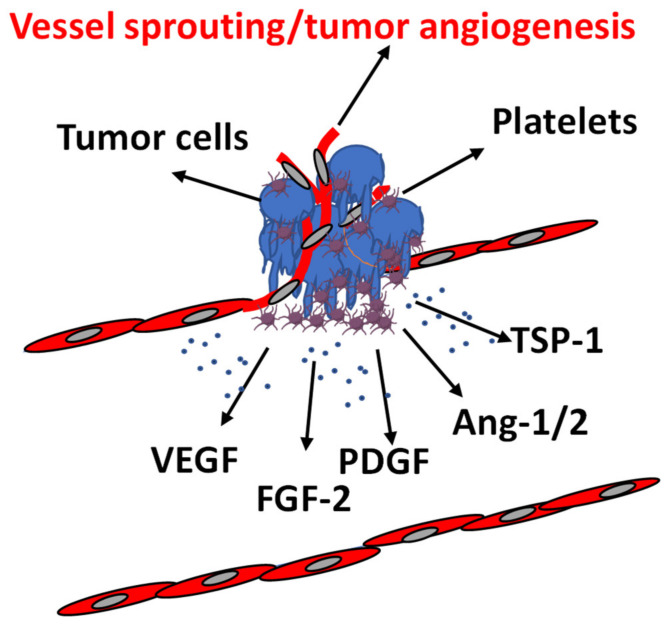
Association of platelets with tumor cells and endothelial cells in TME. Platelets in TME interact with both tumor cells and endothelial cells. Both cells affect platelets releasate and behavior. In TME, the release of platelet-related angiogenic factors promote tumor angiogenesis. VEGF: vascular endothelial growth factor; FGF-2: fibroblast growth factor-2; PDGF: platelet-derived growth factor; Ang1/2: angiopoietin 1/2; TSP-1: thrombospondin-1.

**Table 1 ijms-23-13401-t001:** Preclinical evidence of platelets involvement in tumor progression, mediated by angiogenesis, on in vitro models.

References	Angiogenic Model	Tumor Cells	Stimulus/i	Effects on Angiogenesis	Effects on Tumor Cells
Wu et al., 2015; Wang et al., 2018; [69,70]	Endothelial cell tube formation	Lung cancer cells (A549, H1299, H1975)	Platelet releasate	Increase of 25–50%	Proliferation increased 50–70% Cisplatin sensitivity reduced by 50–70%
Yao et al., 2014; [71]			Platelet-derived cytokines	Increase of 50–70%	
Li et al., 2021; [72]			Platelet releasate	Increase of 25%	
Jiang et al., 2017; [73]		Breast cancer cells (MCF-7, MDA-MB-231)	PAR1/4-induced platelet releasate	Increase of 25%	Proliferation increased by 50–70%
Erices et al., 2017; [74]		Ovarian cancer cells (SKOV3, UCI101)	Platelet with/without metformin	Increase of 25–50%	Migration increased by 25%; Invasiveness increased by 25–30%
Di Vito et al., 2017; [75]		GBM cells	Platelet releasate	Increase of 50–75%	

**Table 2 ijms-23-13401-t002:** Preclinical evidence of the correlation of platelets with MVD and tumor progression in mouse models.

	Tumor Model/Cells	Animal Model	Platelets	Tumor Volume	Tumor Weight	Vessel Density	Angiogenic Factors
Li et al., 2014; [76]	B16-F10 murine melanoma	C57BL/6J black mice or BALB/c white mice	Depletion	No differences	No differences	Decreased MVD	Decreased Ang-1 and increased Ang-2
Zaslavsky et al., 2010; [77]	Lewis lung carcinoma and B16F10 murine melanoma	C57BL/6J black mice	Tsp-1^−/−^	Increase	NR	Decreased MVD	NR
Stone et al., 2012; [78]	Epithelial ovarian cancer	White mice	Depletion	NR	Decrease	Decreased MVD	NR
Feng et al., 2011; [54]	B16-F10 murine melanoma	C57BL/6J black mice	Infusion	NR	NR	Increased MVD	NR
			Depletion	NR	NR	Decreased MVD	NR
Yuan et al., 2015; [79]	SKOV3 Human ovarian cancer cells	White mice	Infusion	Increase	NR	Increased MVD	Increased VEGF
			Depletion	Decrease	NR		
Martini et al., 2020; [80]	B16-F10 murine melanoma	Bcl-x Ptl20/Ptl20 black mice on C57BL/6 background	Decrease of 25%	No differences	No differences	Increased MVD	NR
Brockmann et al., 2011 [81]	G55T2 glioblastoma cells	NMRI-nu/nu white mice	Depletion	No differences	NR	No differences	NR

NR: not reported; MVD: Microvessel density; Ang1/2: angiopoietin-1/2.

**Table 3 ijms-23-13401-t003:** Clinical evidence of platelet involvement on tumor angiogenesis.

Ref	Tumor Type	Number of Patients	Stage	PLTs				PLTs Markers				PMP
				N°	MPV	VEGF (pg/10^6^ plts)	IL-6 (pg/mL)	PF4 (ng/10^6^ plts)	TSP-1 (ng/10^6^ plts)	TGFBβ (ng/10^6^ plts)	PDGF (ng/10^6^ plts)	
Chater et al., 2018; [82]	Colorectal cancer	Pt. 30		259 × 10^9^/L		15.55 pg/actin unit					304 pg/actin unit	
		Ctr. 30		246 × 10^9^/L		2.59 pg/actin unit					125 pg/actin unit	
Di Vito et al., 2016; [75]	Glioblastoma	Pt. 22		Low VEGF: 238 nL High VEGF: 248 nL		41.05 (low) 82.66 (high)						
Han et al., 2014; [83]	Breast cancer	Pt. 37	I/II: Pt 24 III/IV: Pt 13			Stage I/II 2.37 Stage III/IV 3.38		Stage I/II 21.6 Stage III/IV 24.4	29.2	Stage I/II 16.5 Stage III/IV 24.6	Stage I/II 40.4 pg/10^6^ plts Stage III/IV 49.9 pg/10^6^ plts	
		Ctr. 65				0.9		10.2	27.0	Median 4.3	Median 19.1 pg/10^6^ plts	
Kim et al., 2003; [84]	Gastric cancer	Pt. 109	I: Pt 28 II/III: Pt 40 IV: Pt 41	Stage I 234 × 10^9^/mL Stage II/III 227 × 10^9^/mL Stage IV 274 × 10^9^/mL		Plasmatic level Stage I 23.5 pg/mL Stage II/III 24.4 pg/mL Stage IV 30.2 pg/mL	Stage I 3.40 Stage II/III 3.99 Stage IV 6.00					Stage I 0.76 × 10^9^/mL Stage II/III 0.87 × 10^9^/mL Stage IV 7.16 × 10^9^/mL
		Ctr. 29				Plasmatic level 24.6 pg/mL	2.95					0.42 × 10^9^/mL
Mayer et al., 2012; [85]	Breast cancer	Pt. 23				≈1400 ng/mL		≈600 pg/mL				
		Ctr. NR				≈100 ng/mL		≈200 pg/mL				
McDowell et al., 2005; [86]	Breast cancer	Pt. 11				Thrombin stimulation: 1.423 TF stimulation: 1.045 Time exposure: 0.104			Thrombin stimulation: 0.218 TF stimulation: 0.045 Time exposure: 0.030			
		Ctr. 11				Thrombin stimulation: 0.476 TF stimulation: 0.510 Time exposure: 0.017			Thrombin stimulation: 0.126 TF stimulation: 0.030.1 Time exposure: 0.070			
Sabrkhany et al., 2017; [87]	Lung cancer	Pt. 86	I/II: Pt 39 III/IV: Pt 47	Stage I/II ≈ 225 × 10^9^/L Stage III/IV ≈ 310 × 10^9^/L	Stage I/II ≈ 7.0 fL Stage III/IV ≈ 6.0 fL	Stage I/II ≈ 0.8 Stage III/IV ≈ 0.75		Stage I/II ≈ 38 Stage III/IV ≈ 26	Stage I/II ≈ 63 Stage III/IV ≈ 55		Stage I/II ≈ 37 Stage III/IV ≈ 46	
		Ctr. NR		≈210 × 10^9^/L	≈6.5 fL	≈0.4		≈36	≈62		≈36	
	Pancreatic cancer	Pt.42	I/II: Pt 29 III/IV: Pt 13	Stage I/II ≈ 275 × 10^9^/L Stage III/IV ≈ 260 × 10^9^/L	Stage I/II ≈ 7.1 fL Stage III/IV ≈ 7.5 fL	Stage I/II ≈ 0.7 Stage III/IV ≈ 0.6						
		Ctr. NR		≈225 × 10^9^/L	≈6.5 fL	≈0.3						
Seo et al., 2010; [88]	Gastric cancer	Pt.148	I/II: Pt 21 III: Pt 29 IV: Pt 98	240 × 10^6^/mL		Stage I/II: 1.16 Stage III: 2.02 Stage IV: 2						
Tokyol et al., 2009; [89]	Breast cancer	Pt. 20		281.359 ± 17.12 (1000/mm^3^)								
		Ctr. 14		236.79 ± 12.84 (10,000/mm^3^)								
Yao et al., 2014; [71]	Lung cancer (NSCLC)	Pt.68	I/II: Pt 38 III/IV: Pt 30			Stage I/II: 20.2 Stage III/IV: 60.86			Stage I/II 15.1 Stage III/IV 34.5			
		Ctr.68				NR						
Nafady et al., 2018; [90]	Chronic myeloid leukemia	Pt.60 (30 A + 30 B)		Group A: 391.50 × 10^9^/L Group B: 250.11 × 10^9^/L								Group A 8.67 ± 2.88 Group B 42.5 ± 2.82

PLTs: platelets; Pt: patients; Ctr: controls; MPV: mean platelet volume; VEGF: vascular endothelial growth factor; IL-6 interleukin-6; PF4: platelet factor 4; TSP-1: thrombospondin-1; TGFβ: transforming growth factor beta; PDGF: platelet-derived growth factor; PMP: platelet microparticles; NR: not reported.

## Data Availability

Not applicable.

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
