# Peer review of "Scoping Review on Platelets and Tumor Angiogenesis: Do We Need More Evidence or Better Analysis?"

_ijms, 2022, doi:10.3390/ijms232113401_

Round 1
Reviewer 1 Report
Although there have been many review articles associating platelets with tumor angiogenesis, few to my knowledge have created search criteria to combine in vitro, pre-clinical, and clinical studies to provide a broad assessment of the literature. This article, therefore, adds some novelty compared to previous reviews. The selection criteria were well designed and comprehensive, although more critique on the contributions of platelets towards tumor angiogenesis would provide greater scientific rigor to the manuscript.
Major concerns
1. Overdependence on review articles: References in the introduction section are almost exclusively other review articles (26/34) and do not provide sufficient validation for statements that remain controversial. For example, the authors make definitive statements including “platelets regulate tumor angiogenesis through the release of circulating factors” (Ref 18). If so, the contribution of platelets towards tumor angiogenesis would be dogma and not require a critical review.
2. More critique: Given the unbiased title and abstract, it was surprising to see a lack of counter-evidence where platelets do not contribute to tumor angiogenesis (in vitro, pre-clinical, clinical). The authors should clearly acknowledge this body of work if it exists.
3. Interpretation of experiments: The authors need to acknowledge the limitations of capillary tube formation and how this is not synonymous with angiogenesis. These experiments are isolated (e.g. platelets/platelet releasate and ECs) they thus do not reflect the complexity of the tumor microenvironment.
4. The variable effects of platelets on tumor growth: Given that angiogenesis is essential for tumor growth, the authors need to address the substantial pre-clinical evidence that lowering platelet counts or inhibiting platelet activation does not always limit tumor growth.
5. Discussion on antiplatelet drugs: Many antiplatelet drugs (e.g., aspirin, P2Y12 antagonists) limit platelet activation and the release of angiogenic regulators. The authors should include in their discussion whether limiting platelet activation reduces circulating angiogenic regulators and tumor progression.
6. Platelet levels of angiogenic factors: The fact that platelets are a reservoir for angiogenic factors and some platelet angiogenic proteins (e.g., VEGF) increase in cancer patients alone are not evidence that platelets directly contribute to tumor angiogenesis and may be an artifact (or biomarker) of malignancy. The authors need to make this clear when they provide a more critical discussion of the literature.
7. Mechanisms beyond the release of angiogenic proteins: Some of the clearest evidence that platelets contribute to tumor angiogenesis comes from their effects on the stability of neovasculature. Although this could in part be due to angiogenic factors (e.g., Angiopoietin-1), it can also stem from platelet binding/adhesion through ligand-receptor interactions. The authors should acknowledge that platelets likely contribute to angiogenesis through additional mechanisms, not just the release of angiogenic proteins.
8. Less focus on tumor cell proliferation: although in vitro studies assessing capillary tube formation frequently investigate tumor cell proliferation, the authors should discuss the latter less given the review series is on the tumor microenvironment (i.e., non-malignant cells) and the review article is on angiogenesis.
Minor concerns
1. Although many relevant findings have occurred in the last 20 years, the authors should consider introducing the rationale and hypothesis behind platelets contributing to tumor angiogenesis by discussing the work of Drs. Pinedo and Folkman in 1998.
2. Given the topic of the review series, the article would benefit from identifying which tumors are most reliant on platelet-derived angiogenesis (e.g., ovarian cancer) and comparing their microenvironment with those less responsive to platelets (e.g., breast cancer).
3. The authors generalize platelet “tumor-education” and “activation”. “Tumor-educated platelets” often refer to altered content (e.g., protein, RNA) due to reprogramming of megakaryocytes or circulating platelets by the primary tumor (e.g., alternative splicing, see publications by Best et al.). It does not refer to tumor-induced platelet activation. The authors should correct this.
4. In their introduction, the authors cite that “angiogenic factors are stored in distinct alpha granules”. This has not been definitively proven and is the subject of much debate, with other groups producing conflicting findings. The authors should acknowledge this.
5. Can the authors clarify why in their clinical screening they only selected immune checkpoint inhibitors and no other therapies when used in combination with anti-angiogenic drugs?
6. When discussing the effects of platelet releasate on capillary tube formation, the authors should discuss how the releasate was generated, as several platelet agonists (e.g., ADP) directly induce tube formation.
7. References 52 and 37 should not be included in the clinical section of the manuscript, as they involve taking platelets from patients to manipulate tube formation by endothelial cells. This is an example of in vitro/ex vivo data and although interesting, should be moved to the pre-clinical section.
8. The authors state in their discussion that “the role of the lung as a reservoir of platelets may justify the greater attention to lung cancer models”. The authors should explain what they mean by a reservoir. Are platelets activated/sequestered there? Or are they talking about platelet production by lung megakaryocytes? Also, the pre-clinical lung cancer models described/referenced appear to be from the subcutaneous implantation of lung tumor cells, so would not recreate the lung microenvironment.
Author Response
Please see the attachment
Sincerely
Sandra Donnini

Reviewer 2 Report
In this review manuscript, the authors discussed the literature on platelets and angiogenesis from the past ~20 years. They summarize the state-of-the-art literature in a structured manner, following which they draw their conclusions and initiate the discussion. The manuscript is well written, clear, complete, and provides the readers a solid overview on the available work. I have some minor comments.
- The introduction is rather long. Here, a schematic figure may help the reader to capture the information more easily.
- The discussion section may benefit from a more future-oriented scope. What lacks here is an answer provided to the question included in the manuscripts title (‘do we need more evidence or better analysis?’). Also, what would the authors propose to investigate next? How easily and reliably can preclinical study results be translated to the in vivo setting? What models are required to study the role of platelets in angiogenesis? What is known about platelets in anti-angiogenesis drugs such as bevacizumab? What is the role of platelet subpopulations and anti-platelet drugs on angiogenesis?
- The abstract closes with a ‘cliffhanger’ of which the final sentence is unclear to me (‘or a determinant of an algorithm for analysing tumor angiogenesis’? What algorithm?). The abstract may become more attractive and interpretable by including the conclusion from the review in the final sentence, thereby also allow readers to only study the abstract instead of reading the paper in full.
- There are some typo’s throughout the manuscript text, please double check for this.
Author Response
Please see the attachment.
Sincerely
Sandra Donnini

Round 2
Reviewer 1 Report
The revised version of the manuscript provided by the authors is much improved. It provides a more balanced introduction/discussion section that includes the counter-evidence where platelets may not contribute to tumor angiogenesis in cancer patients. In addition, the authors now conclude their work by addressing their key question, i.e., do we need more evidence or better analysis? while offering insights for future studies. The authors have addressed most of my concerns and I have provided only a few select comments for their revised manuscript.
Major concerns
1. In line 496-498, the authors state that “clinical data have demonstrated the antiangiogenic impact” of antiplatelet therapies like aspirin and P2Y12 antagonists. However, these trials typically assess incidence and survival outcomes in patients and therefore do not directly assess tumor angiogenesis. Furthermore, there is just as much evidence that antiplatelet agents (particularly aspirin) have no or even a detrimental effect on tumor incidence and patient outcomes, highlighted by the recent ASPREE study https://pubmed.ncbi.nlm.nih.gov/32778876/. Therefore, the authors must conduct a more balanced argument for the effects of antiplatelet drugs on patient outcomes.
Minor concerns
1. In line 115, the authors state PF4 is “specifically” expressed in platelets. Whereas the majority of PF4 is platelet/megakaryocyte-derived, it is also stored and released by subtypes of myeloid cells. The authors should therefore change their language from “specifically” to “selectively”.
2. I suggest not starting the final paragraph of your discussion section with “in conclusion” when this is also used at the start of the conclusion section after it.
Author Response

(The authors gave the same response as above.)
